# Differences in the performance of resuscitation according to the resuscitation guideline terminology during infant cardiopulmonary resuscitation: "Approximately 4 cm" versus "at least one-third the anterior-posterior diameter of the chest"

**Wongyu Lee, Dongjun Yang, Je Hyeok Oh**[ORCID]*

Department of Emergency Medicine, Chung-Ang University College of Medicine, Seoul, Republic of Korea

* jehyeokoh@cau.ac.kr

## Abstract

### Aim

This study was conducted to investigate the effect of resuscitation guideline terminology on the performance of infant cardiopulmonary resuscitation (CPR).

### Methods

A total of 40 intern or resident physicians conducted 2-min CPR with the two-finger technique (TFT) and two-thumb technique (TT) on a simulated infant cardiac arrest model with a 1-day interval. They were randomly assigned to Group A or B. The participants of Group A conducted CPR with the chest compression depth (CCD) target of "approximately 4 cm" and those of Group B conducted CPR with the CCD target of "at least one-third the anterior-posterior diameter of the chest". Single rescuer CPR was performed with a 15:2 compression to ventilation ratio on the floor.

### Results

In both chest compression techniques, the average CCD of Group B was significantly deeper than that of Group A (TFT: 41.0 [range, 39.3–42.0] mm vs. 36.5 [34.0–37.9] mm, $P = 0.002$; TT: 42.0 [42.0–43.0] mm vs. 37.0 [35.3–38.0] mm, $P < 0.001$). Adequacy of CCD also showed similar results (Group B vs. A; TFT: 99% [82–100%] vs. 29% [12–58%], $P = 0.001$; TT: 100% [100–100%] vs. 28% [8–53%], $P < 0.001$).

### Conclusions

Using the CCD target of "at least one-third the anterior-posterior diameter of the chest" resulted in deep and adequate chest compressions during simulated infant CPR in contrast

**Data Availability Statement:** All relevant data are within the manuscript and its Supporting Information files.

**Funding:** The author(s) received no specific funding for this work.

**Competing interests:** The authors have declared that no competing interests exist.

to the CCD target of "approximately 4 cm". Therefore, changes in the terminology used in the guidelines should be considered to improve the quality of CPR.

## Trial registration

Clinical Research Information Service; cris.nih.go.kr/cris/en (Registration number: KCT0003486).

## Introduction

Adequate chest compression depth (CCD) is one of the key components for improving survival rate of out-of-hospital cardiac arrest [1–3]. Recommended CCD for adult cardiac arrest patients had been adjusted from "at least 5 cm" to "approximately 5 cm" in the 2015 international consensus of International Liaison Committee on Resuscitation because Stiell et al. reported that maximum survival was expected in the depth interval from 40.3 to 55.3 mm (peak, 45.6 mm) [4–6]. These changes had influenced international guidelines [1–3, 7]. Although the 2015 American Heart Association and European Resuscitation Council guidelines maintained the recommended CCD as "at least 5 cm", the 2015 Resuscitation Council of Asia and Korean guidelines for CPR changed the recommended CCD to "approximately 5 cm" according to the 2015 international consensus [3, 4, 7]. This discrepancy among the CPR guidelines confused both the instructors and trainees of CPR. Additionally, the word "approximately" might limit the chance of achieving adequate CCD. Indeed, Trethewey et al. reported recently that the performance of CCD decreased when the recommended CCD included the word "approximately" in contrast to the performance seen with "at least" during adult cardiac arrest simulation [8].

The CPR guidelines for infants recommend that the CCD should be "approximately 4 cm" or "at least one-third the anterior-posterior diameter of the chest" simultaneously [9–11]. We hypothesized that the CCD achieved following the CCD target of "approximately 4 cm" might be lower than CCD achieved following the CCD target "at least one-third the anterior-posterior diameter of the chest". However, there was a possibility that the different terminologies for the CCD might affect other CPR performance variables such as hand position, chest compression rate, chest wall recoil, hands-off time, and ventilation parameters. Therefore, we also examined these parameters as secondary outcomes based on the different CCD terminologies used.

## Materials and methods

### Study ethics

The study protocol and informed consent form were approved by the Chung-Ang University Hospital Institutional Review Board on 3rd December 2018 (Approval number: 1803-012-347). The study was registered at the Clinical Research Information Service (cris.nih.go.kr/cris/en) on 11th February 2019 (Registration number: KCT0003486). The first participant was recruited on 22th February 2019 and the last on 17th August 2019. All participants provided written informed consent before participating in the study.

### Study design

The present study was a prospective randomized simulation trial with parallel arms (Fig 1).

All participants conducted a 2-min CPR with the two-finger chest compression technique (TFT) and two-thumb encircling hands technique (TT) on a simulated infant cardiac arrest model. A wash out period of 1 day was provided between the experiments. Further, the study

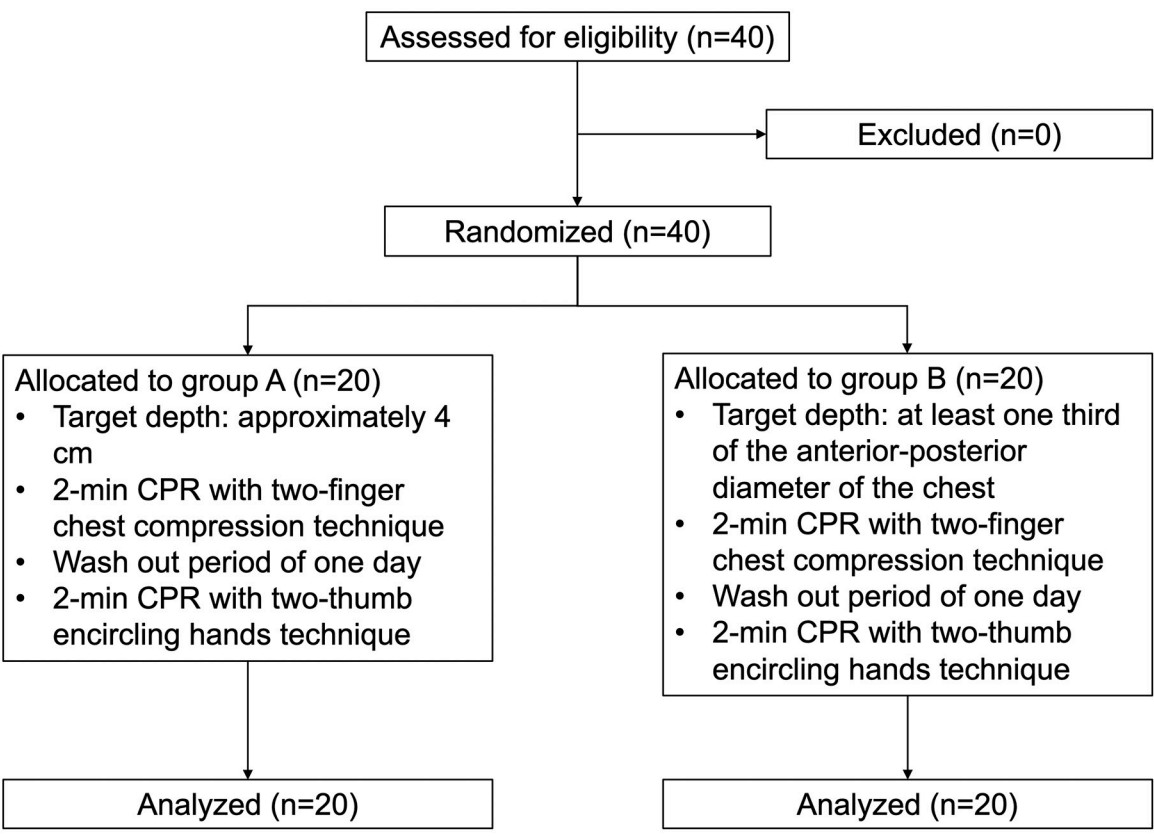

**Fig 1. Study flow diagram.** CPR, cardiopulmonary resuscitation.

participants were randomized to Group A or B with a ratio of 1:1. Participants of Group A conducted all experiments with the CCD target of "approximately 4 cm" and those of Group B conducted all experiments with the CCD target of "at least one-third the anterior-posterior diameter of the chest". Data were analysed after completion of all experiments.

## Study participants

The inclusion criteria were as follows: intern or resident physicians who completed the basic life support course for healthcare professionals within 2 years in our hospital and decided to participate in the study voluntarily after receiving a complete description of the study protocol. Age, sex, and race were not considered as limitations for study participation. The exclusion criteria were as follows: doctors who could not conduct infant CPR because of injuries sustained to their hands or fingers within the last 2 weeks; doctors who declined participation in the study or withdrew their consent within the study period; and doctors who injured their hands or fingers during the experiments.

## Study protocol

The experiments were conducted with an infant cardiac arrest simulation model involving a three-month-old-sized infant manikin (Resusci Baby QCPR, Laerdal Medical, Stavanger, Norway). The same manikin was used in all experiments. The anterior-posterior diameter of the chest of the manikin was 12 cm. The manikin was laid on the floor in the supine position. All experiments were performed with a CPR duration of 2 minutes, and with a chest

compression-to-ventilation ratio of 15:2. Using a face shield, mouth-to-mouth ventilation was administered. The researcher who was in charge of data collection instructed the participants to use the index and middle fingers of the dominant hand when they used TFT for chest compressions, because the index and middle fingers (of the dominant hand) in combination are the most effective, second only to the two-thumb technique [12, 13]. Additionally, for all experiments, the position of the participants was standardized as follows: the right-handed participants were asked to sit on the right side of the manikin and the left-handed participants were asked to sit on the left side of the manikin in the kneeling position. The non-dominant hand was used to maintain head-tilt position during the chest compression period for TFT. Data were collected during the experiments by a sensor embedded in the manikin and were extracted through a computer (Simpad SkillReporter, Laerdal Medical). The Simpad screen was hidden and not visible to either the participant or the researcher.

## Randomization method

The study participants were allocated randomly to Group A or B with stratified randomization method according to their sex for the purpose of allocating the participants in each group with the same ratio of sex. We controlled for sex based on the following reasons: the CCD for women rescuers was inferior to those of men rescuers because factors such as lower body weight and low level of muscular fitness could decrease CCD [14–17]. The present study was a randomized controlled trial that included minimal number of participants. Therefore, a difference in the sex ratio between the two groups might act as a potential confounder. As a result, two randomization lists were created through assignment of a random number (from 1 to 6) to six permuted blocks (1: AABB, 2: BBAA, 3: ABAB, 4: BABA, 5: ABBA, 6: BAAB) obtained through a computer program, with the initial letter of each group being either "A" or "B".

## Outcome variables

The primary outcome variable was average compression depth (ACD, mm). Secondary outcome variables were the ratio of correct hand position around lower half of sternum (correct hand position, %), total compressions (n), the ratio of chest compressions achieving chest compression depth over 38 mm (adequate depth, %), the ratio of complete chest wall recoil within 5 mm from baseline (complete recoil, %), average compression rate (n/min), the ratio of adequate chest compression rate between 100/min to 120/min (adequate rate, %), hands-off time (sec), total ventilations (n), and average ventilation volume (ml). Moreover, data on the characteristics of study participants, such as sex, age, position at the hospital, and dominant hand, were collected.

## Sample size calculation

The ACD for TFT with the index and middle fingers of the right hand was 41.6 ± 1.1 mm and the ACD for TT was 43.1 ± 0.9 mm in a previous study [12]. We hypothesized that the ACD with a target CCD of "approximately 4 cm" would be lower than that with a target CCD of "at least one-third the anterior-posterior diameter of the chest" by as much as the standard deviation (1.1 mm for TFT and 0.9 mm for TT). The two-sided significance level was set at 0.05 and the statistical power was set at 80%. With a drop-out rate of 20%, the minimum number of study participants in each group was calculated to be 20 using a web-based program (Sample size calculator: two parallel-sample means). Thus, 40 participants were finally recruited.

## Statistical analysis

All statistical analyses were performed using IBM SPSS Statistics, version 25.0 (IBM Corp., Armonk, New York, USA). Continuous variables were presented as mean ± standard deviation or median (interquartile range) according to the normality of the data distribution, whereas categorical variables were expressed as frequencies and percentages. The normality of the data distribution was analysed using the Shapiro-Wilk test or the Kolmogorov-Smirnov test. For normally distributed data, the two-sided independent sample *t*-test was used for the comparison of data between groups, otherwise, the Mann-Whitney *U* test was used. The difference between the categorical variables was assessed with the Pearson's chi-squared test or Fisher's exact test as appropriate. A *P*-value of less than 0.05 was considered statistically significant.

# Results

## Baseline characteristics

Overall, 40 intern or resident physicians were recruited and assigned randomly to the two groups (Group A = 20, Group B = 20). None of the participants were excluded. There were no significant differences in the sex, age, position at the hospital, specialities of the resident physician, and handedness between Groups A and B (Table 1).

## Comparisons between Group A and B

The ACDs of Group B were significantly deeper than those of Group A for both chest compression techniques (TFT: 41.0 [39.3–42.0] mm vs. 36.5 [34.0–37.8] mm, *P* = 0.002; TT: 42.0 [42.0–43.0] mm vs. 37.0 [35.3–38.0] mm, *P* < 0.001; Table 2, Fig 2). The proportion of trials wherein adequate depth was achieved was also significantly higher in Group B (TFT: 99% [82–100%] vs. 29% [12–58%], *P* = 0.001; TT: 100% [100–100%] vs. 28% [8–53%], *P* < 0.001). The proportion of trials where complete recoil was noted in Group B was significantly higher than that in Group A for TFT (98% [88–100%] vs. 87% [76–92%], *P* = 0.001). However, no significant difference in this regard was noted for TT (Table 2). Other variables, including correct

**Table 1. Characteristics of the study participant according to the study group.**

| Variables | Group A (n = 20) | Group B (n = 20) | *P*-value |
|---|---|---|---|
| Male sex (%) | 15 (75.0) | 15 (75.0) | 1.000 |
| Age (years) | 27.5 (26.0, 29.0) | 26.5 (25.0, 29.5) | 0.529 |
| Position | | | 1.000 |
| Resident physician | 11 (55.0) | 11 (55.0) | |
| Intern physician | 9 (45.0) | 9 (45.0) | |
| Specialities of the resident physician | | | 0.830 |
| Emergency Medicine | 3 (27.3) | 4 (36.4) | |
| Internal Medicine | 4 (36.4) | 3 (27.3) | |
| Pediatrics | 2 (18.2) | 0 (0.0) | |
| General Surgery | 0 (0.0) | 2 (18.2) | |
| Orthopedics | 0 (0.0) | 1 (9.1) | |
| Dermatology | 1 (9.1) | 1 (9.1) | |
| Psychiatrics | 1 (9.1) | 0 (0.0) | |
| Right handedness (%) | 18 (90.0) | 19 (95.0) | 1.000 |

Data are expressed as n (%) except for age, which is presented as median and interquartile range.

**Table 2. Comparisons of the variables according to the group.**

| Variables | Group A (n = 20) | Group B (n = 20) | *P*-value |
|---|---|---|---|
| **Infant cardiopulmonary resuscitation using the two-finger chest compression technique** | | | |
| Correct hand position (%) | 100 (100, 100) | 100 (100, 100) | 0.779[†] |
| Total compressions (n) | 136 (129, 145) | 144 (134, 150) | 0.192[†] |
| Average compression depth (mm) | 36.5 (34.0, 37.8) | 41.0 (39.3, 42.0) | **0.002[†]** |
| Adequate depth (%) | 29 (12, 58) | 99 (82, 100) | **0.001[†]** |
| Complete recoil (%) | 87 (76, 92) | 98 (88, 100) | **0.001[†]** |
| Average compression rate (n/min) | 120 ± 14 | 123 ± 16 | 0.480[*] |
| Adequate rate (%) | 22 (2, 82) | 6 (0, 45) | 0.289[†] |
| Hands-off time (sec) | 52 (50, 55) | 50 (49, 54) | 0.211[†] |
| Total ventilations (n) | 16 (14, 18) | 16 (14, 18) | 0.659[†] |
| Average volume (ml) | 41 (36, 44) | 38 (32, 42) | 0.383[†] |
| **Infant cardiopulmonary resuscitation using the two-thumb encircling hands technique** | | | |
| Correct hand position (%) | 100 (100, 100) | 100 (100, 100) | 0.799[†] |
| Total compressions (n) | 136 (135, 149) | 136 (135, 150) | 0.820[†] |
| Average compression depth (mm) | 37.0 (35.3, 38.0) | 42.0 (42.0, 43.0) | **<0.001[†]** |
| Adequate depth (%) | 28 (8, 53) | 100 (100, 100) | **<0.001[†]** |
| Complete recoil (%) | 12 (9, 19) | 11 (9, 21) | 0.698[†] |
| Average compression rate (n/min) | 131 (129, 134) | 130 (127, 140) | 0.779[†] |
| Adequate rate (%) | 1 (0, 9) | 2 (0, 5) | 0.242[†] |
| Hands-off time (sec) | 53 ± 5 | 54 ± 4 | 0.888[*] |
| Total ventilations (n) | 15 ± 3 | 16 ± 4 | 0.576[*] |
| Average volume (ml) | 47 ± 7 | 49 ± 10 | 0.540[*] |

*P* < 0.05 are presented in bold.

Group A: chest compression depth target of "approximately 4 cm"

Group B: chest compression depth target of "at least one-third the anterior-posterior diameter of the chest"

[*]Statistical significance was tested using two-sided independent sample *t*-test.

[†]Statistical significance was tested using Mann-Whitney *U* test.

hand position, total compressions, average compression rate, adequate rate, hands-off time, total ventilations, and average volume, were not significantly different between the groups.

## Discussion

The results of the present study supported our hypothesis that the CCD according to the CCD target of "at least one-third the anterior-posterior diameter of the chest" was deeper than the CCD according to the CCD target of "approximately 4 cm".

Deakin et al. also reported the performance of CPR according to different CCD targets [18]. Although the rescuers could discriminate the different CCD targets precisely, they did not achieve the target depths in all trials. We speculate that the target depth was not achieved because the depths were presented as ranges (e.g. 4.0–5.0 cm, 4.5–5.5 cm, and 5.0–6.0 cm). The rescuers might limit the force applied because of the upper margin of the target depth range. Considering these results, we hypothesized that if the term "approximately" is used to define the CCD, results similar to those seen when the CCD was defined as a range will be noted.

Previously, the 2005 guidelines recommended the CCD for adults as a range (Table 3) [19–21].

This range-based definition of the recommended depth was subsequently removed from the 2010 guidelines [5, 22, 23]. Simultaneously, the recommended depth for the paediatric

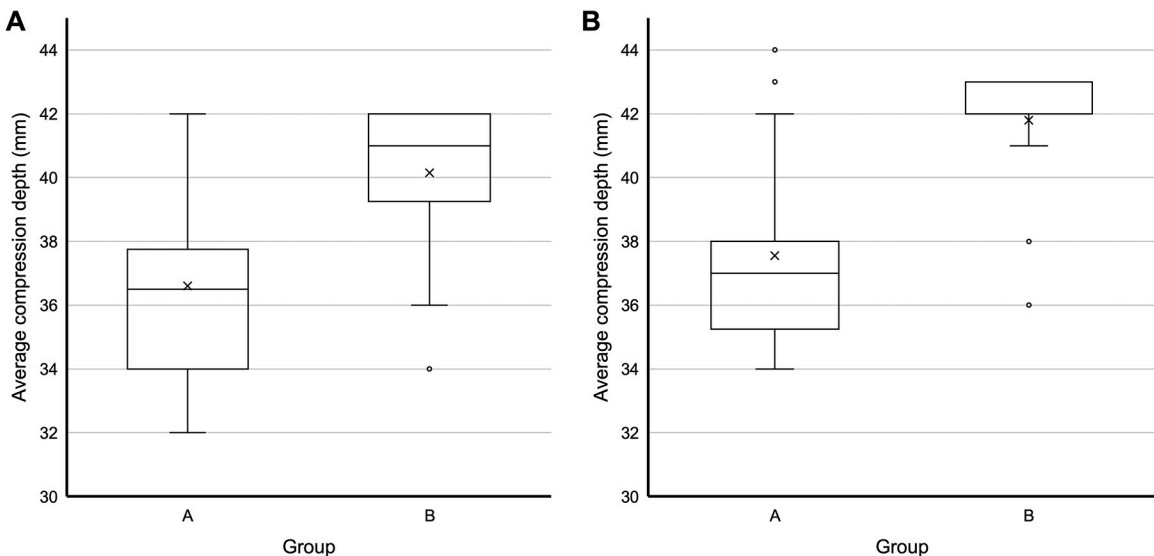

**Fig 2. Comparisons of average compression depths according to the group.** A. Two-finger chest compression technique ($P = 0.002$), B. Two-thumb encircling hands technique ($P < 0.001$).

population was changed from "approximately 1/3 of the chest" to "at least 1/3 of the chest" [21, 24, 25, 27–30]. However, the term of "approximately" had not been completely removed from the guidelines, and "approximately 5 cm" or "approximately 4 cm" has been used since the 2010 guidelines were formulated. Although the International Liaison Committee on Resuscitation and European Resuscitation Council changed "approximately" to "by" in the 2015 paediatric guidelines, the term "approximately" remained in the 2015 American Heart Association

**Table 3. Recommended chest compression depth in the international guidelines since 2005.**

|  | 2005 | 2010 | 2015 |
|---|---|---|---|
| Recommended depth for adults |  |  |  |
| International consensus of ILCOR | At least 4–5 cm [19] | Al least 5 cm [5] | Approximately 5 cm [4] |
| ERC guideline | 4–5 cm [20] | At least 5 cm (but not exceeding 6 cm) [22] | At least 5 cm (but not exceeding 6 cm) [2] |
| AHA guideline | Approximately 4–5 cm [21] | At least 5 cm [23] | At least 5 cm (but not exceeding 6 cm) [1] |
| Recommended depth for child |  |  |  |
| International consensus of ILCOR | Approximately 1/3 of the chest [24] | At least 1/3 of the chest or approximately 5 cm [25] | At least 1/3 of the chest or by 5 cm [26] |
| ERC guideline | Approximately 1/3 of the chest [27] | At least 1/3 of the chest or approximately 5 cm [28] | At least 1/3 of the chest or by 5 cm [9] |
| AHA guideline | Approximately 1/3-1/2 of the chest [21] | At least 1/3 of the chest or approximately 5 cm [29] | At least 1/3 of the chest or approximately 5 cm [10] |
| Recommended depth for infant |  |  |  |
| International consensus of ILCOR | Not evaluated | At least 1/3 of the chest or approximately 4 cm [25] | At least 1/3 of the chest or by 4 cm [26] |
| ERC guideline | Approximately 1/3 of the chest [27] | At least 1/3 of the chest or approximately 4 cm [28] | At least 1/3 of the chest or by 4 cm [9] |
| AHA guideline | Approximately 1/3-1/2 of the chest [21] | At least 1/3 of the chest or approximately 4 cm [29] | At least 1/3 of the chest or approximately 4 cm [10] |

AHA, American Heart Association; ERC, European Resuscitation Council; ILCOR, International Liaison Committee on Resuscitation

guidelines [9, 10, 26]. Moreover, the term "by" might cause confusion in the same manner as with "approximately".

The results of the present study showed that compared to "at least one-third the anterior-posterior diameter of the chest", the term "approximately 4 cm" could significantly limit the CCD. The CCD has been reported to be usually higher with TT when compared to TFT [12, 31–34]. However, in this study, ACD and the proportion of cases wherein the adequate depth was achieved were significantly different between the groups for both TT and TFT (Fig 2).

Another interesting result was regarding the proportion of trials where complete recoil was noted with the TFT (Table 2). This result suggests that compared to the term "at least one-third the anterior-posterior diameter of the chest", the term "approximately 4 cm" could limit the proportion of trials with complete recoil. However, similar results were not noted for TT. In case of TT, the proportion of trials with complete recoil were low in both groups, and these findings are consistent with those of a previous study [35]. However, we could not confirm why the proportion of trials with complete recoil with the TT was low in both the groups.

Previous studies have demonstrated that higher survival is associated with deeper CCD in the paediatric population [36]. Considering the negative (e.g. decreasing CCD) effect of the term "approximately 4 cm", changes in the terminology used in the guidelines should be considered.

This study had several limitations that warrant mention. First, we recruited only medical doctors; therefore, our results cannot be generalised to other types of rescuers. Second, since the results of the present study were obtained from simulation experiments with a manikin, the results may not be identical to those obtained from CPR performed on humans. Third, we only compared two terminologies, i.e. "approximately" and "at least", while "4 cm" and "one-third the anterior-posterior diameter of the chest" are also different. Therefore, there is a possibility that the term "one-third the anterior-posterior diameter of the chest" might have been attributable for the differences noted in the CCD between the groups in contrast to "4 cm". However, the anterior-posterior diameter of the chest of the manikin was 12 cm, and thus, theoretically, "4 cm" and "anterior-posterior diameter of the chest" are the same. Nevertheless, the rescuers might perceive these two terms differently, and consequently, perform CPR in a different manner.

## Conclusions

Using the CCD target of "at least one-third the anterior-posterior diameter of the chest" resulted in deep and adequate chest compressions during infant CPR when compared to with the CCD target of "approximately 4 cm". Therefore, changes in the terminology used in the guidelines should be considered to improve the quality of CPR.

## Supporting information

**S1 Dataset.**
(XLSX)

## Acknowledgments

The authors thank all participants from our hospital for their contribution to this study.

## Author Contributions

**Conceptualization:** Je Hyeok Oh.

**Data curation:** Wongyu Lee, Dongjun Yang.

**Formal analysis:** Je Hyeok Oh.

**Investigation:** Wongyu Lee, Dongjun Yang, Je Hyeok Oh.

**Methodology:** Je Hyeok Oh.

**Project administration:** Je Hyeok Oh.

**Resources:** Je Hyeok Oh.

**Software:** Je Hyeok Oh.

**Supervision:** Je Hyeok Oh.

**Validation:** Je Hyeok Oh.

**Visualization:** Je Hyeok Oh.

**Writing – original draft:** Wongyu Lee, Je Hyeok Oh.

**Writing – review & editing:** Je Hyeok Oh.

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
