## [Decision Letter · Decision Letter 0]

24 Jan 2020

PONE-D-19-34142

Differences in the performance of resuscitation according to the resuscitation guideline terminology during infant cardiopulmonary resuscitation: “approximately 4 cm” versus “at least one-third the anterior-posterior diameter of the chest”

PLOS ONE

Dear Prof. Oh,

Thank you for submitting your manuscript to PLOS ONE. After careful consideration, we feel that it has merit but does not fully meet PLOS ONE’s publication criteria as it currently stands. Therefore, we invite you to submit a revised version of the manuscript that addresses the points raised during the review process.

Please review and address all the feedback below, focusing especially on the comments related to clarity around the outcomes.

We would appreciate receiving your revised manuscript by Mar 09 2020 11:59PM. To enhance the reproducibility of your results, we recommend that if applicable you deposit your laboratory protocols in protocols.io, where a protocol can be assigned its own identifier (DOI) such that it can be cited independently in the future. For instructions see: http://journals.plos.org/plosone/s/submission-guidelines#loc-laboratory-protocols

We look forward to receiving your revised manuscript.

Kind regards,

Kevin Ching, M.D.

Academic Editor

PLOS ONE

Journal Requirements:

Reviewers' comments:

Reviewer's Responses to Questions

**Comments to the Author**

1. Is the manuscript technically sound, and do the data support the conclusions?

Reviewer #1: Partly

Reviewer #2: No

Reviewer #3: Yes

2. Has the statistical analysis been performed appropriately and rigorously? 

Reviewer #1: I Don't Know

Reviewer #2: No

Reviewer #3: Yes

3. Have the authors made all data underlying the findings in their manuscript fully available?

Reviewer #1: Yes

Reviewer #2: Yes

Reviewer #3: Yes

4. Is the manuscript presented in an intelligible fashion and written in standard English?

Reviewer #1: No

Reviewer #2: Yes

Reviewer #3: Yes

5. Review Comments to the Author

Reviewer #1: 1. In line 49, what is being referred to as the “same problem”? Is it the confusion that is detailed in Lines 44-45, or the decrease performance of CCD referred to in lines 45-48, or both?

2. In Line 51, “the quality of CPR” is not explicitly defined as what the study designates to determine what “quality” is/means.

3. The hypothesis, lines 51-55, states the belief that one CCD target will result in “lower” quality of CPR than the other CCD target. As the hypothesis lays out the objective of proving one being lesser than the other, the data, analysis, and discussion is based upon the context of one being better than the other. Although this is one in the same, it would be more consistent to present the information uniformly (i.e. discussing how one CCD target is of lower quality rather than of higher quality vs. stating a hypothesis that one CCD target is of better quality than the other).

4. Under Study Participants, line 74-75 states that “Age, sex, and race were not considered as limitations for study participations.” However, under Randomization Method, lines 104-105, groups were randomized by stratifying according “to their sex for the purpose of allocating the participants in each group with the same ratio of sex.” Is not sex, therefore, an inclusion criterion for study participation if the aim of randomization of the groups were to make sure the sex ratio was equivalent? If no participants were dropped, then was those recruiting participants aware and/or considering the number of males and females that were approached and volunteered for the study?

5. “Quality,” again, has not been defined for this study. Therefore, when discussing quality, as is in Lines 191-192, in comparison to the quality of chest compression in other studies, it becomes difficult for the reader to understand the comparison/contrast being made. Please define what the study indicates as “quality CPR” and “quality chest compressions”

6. Caution when using the term “significantly.” In lines 198-200, when using the phrase “significantly lower” it eludes to the reader that there was a statistically significant (P < 0.05) outcome where in fact the authors are stating the opposite. For example one can say based on the findings of this study that there was no statistical difference between groups A and B for complete recoil and the proportion of trials where complete recoil was noted was fewer than that required for overall high quality CPR, which is consistent to a previous study. Not that, line 198, “was significantly lower than that required for high quality CPR...”

7. Lines 201-202, “…the risk of injury might be increased when using the term “at least” contradicts the study’s conclusions and hypothesis. If stating/concluding Group B, with CCD target of “at least”, was better than Group A, CCD target of “approximately”, then by having Lines 201-202 causes doubt and conflicting view point. Be careful on wording in addressing this point.

8. Based on the third limitation outlined in Lines 208-214, the issue in the discussion and conclusion is that one cannot state that the use of “approximate” vs “at least” being better than the other, as the author has done, without accounting for confounding variables which is the entire phrase of the CCD target. One cannot conclude, based on the study design and data presented, that the differences between the groups were to “approximate” vs “at least” or “one third the anterior-posterior diameter of the chest” vs “4 cm.” It can only be discussed/concluded on the entire phrase itself and not to the segments within the phrase. Would therefore remove the context around Lines 190-191 and Lines 195-197.

Reviewer #2: PONE-D-19-34142: statistical review

SUMMARY. This study estimates the effect of two resuscitation guidelines on the performance of infant cardiopulmonary resuscitation, measured in terms of average compression depth (ACD; primary outcome) and a battery of secondary outcomes. I have some concerns about the definition of the variables under study (major issue 1), the presentation of the results (major issue 2) and, finally, a possible limitation of the study (major issue 3)

MAJOR ISSUES

1. Lines 110-115. The description of the secondary outcomes is a bit cryptic. For example, how is "correct hand position" defined? What does the number of total compressions measure? When is depth "adequate"? When is recoil "complete"? A detailed description of the outcomes under study is not only a courtesy to the readers (such as me) who are not necessarily experts of resuscitation methods, but it also allows results reproducibility. In addition, in which sense do these variables measure resuscitation performance?

2. In the statistical analysis section, the authors correctly say that they use the Chi-square tests or Fisher exact tests to compare proportions. However, in Table 2 differences between proportions are incorrectly tested using the Mann-Whitney U test (at least this is what the note of the table says). Please clarify.

3. Lines 89-90: the same manikin was used in all the experiments. Isn't this a limitation of the study? Shouldn't one exploit manikins of different sizes to robustify the results? Please clarify this point.

Reviewer #3: The methodology appears to be sound, and the results do appear to be consistent with the data (supplemental data).

Their statistical analysis plan appears to be appropriate as well. Though I wish they would have described or mentioned what data was normally distributed or not.

It appears that it was only interns or residents that participated in the study. I would have mentioned this in the study, instead of saying medical doctors.

6. PLOS authors have the option to publish the peer review history of their article (what does this mean?). If published, this will include your full peer review and any attached files.

Reviewer #1: Yes: Adnan Mesiwala, DO

Reviewer #2: No

Reviewer #3: No

---

## [Author Response · Author response to Decision Letter 0]

30 Jan 2020

30 January 2020

Kevin Ching, MD

Academic Editor

PLOS ONE

Dear Editor, 

I wish to re-submit the manuscript titled “Differences in the performance of resuscitation according to the resuscitation guideline terminology during infant cardiopulmonary resuscitation: “approximately 4 cm” versus “at least one-third the anterior-posterior diameter of the chest”.” The manuscript ID is PONE-D-19-34142.

We thank you and the reviewers for your thoughtful suggestions and insights. The manuscript has benefited from these insightful suggestions. I look forward to working with you and the reviewers to move this manuscript closer to publication in PLOS ONE. The manuscript has been rechecked and the necessary changes have been made in accordance with the reviewers’ suggestions. Thank you for your consideration. I look forward to hearing from you.

Sincerely,

Je Hyeok Oh, MD, PhD

Department of Emergency Medicine,

Chung-Ang University College of Medicine,

84 Heukseok-ro, Dongjak-gu, Seoul 06974, Republic of Korea

Tel. no.: +82 2 6299 1820

Fax no.: +82 2 6264 1119

Email address: jehyeokoh@cau.ac.kr

 

Our response to the comments of the Reviewer #1:

1. In line 49, what is being referred to as the “same problem”? Is it the confusion that is detailed in Lines 44-45, or the decrease performance of CCD referred to in lines 45-48, or both?

Answer) Thank you for your comments. The problem refers to the fact that the guideline of cardiopulmonary resuscitation used both terminologies “at least” and “approximately” simultaneously. We have deleted the sentence to avoid misinterpretation.

2. In Line 51, “the quality of CPR” is not explicitly defined as what the study designates to determine what “quality” is/means.

Answer) Thank you for your comments. Our study focused on whether the different terminologies might affect the chest compression depth. Therefore, we changed the phrase “the quality of CPR” to “the chest compression depth” to clarify the hypothesis.

3. The hypothesis, lines 51-55, states the belief that one CCD target will result in “lower” quality of CPR than the other CCD target. As the hypothesis lays out the objective of proving one being lesser than the other, the data, analysis, and discussion is based upon the context of one being better than the other. Although this is one in the same, it would be more consistent to present the information uniformly (i.e. discussing how one CCD target is of lower quality rather than of higher quality vs. stating a hypothesis that one CCD target is of better quality than the other).

Answer) Thank you for your comments. The primary goal of the present study was to compare the chest compression depth (CCD) achieved based on the different terminologies used for defining the CCD. Therefore, we arranged the text of the manuscript so that it focuses on the CCD. However, there was a possibility that the different terminologies for the CCD might affect other CPR performance variables such as hand position, chest compression rate, chest wall recoil, hands-off time, and ventilation parameters. Therefore, we also examined these variables as secondary outcome based on the different CCD terminologies used.

4. Under Study Participants, line 74-75 states that “Age, sex, and race were not considered as limitations for study participations.” However, under Randomization Method, lines 104-105, groups were randomized by stratifying according “to their sex for the purpose of allocating the participants in each group with the same ratio of sex.” Is not sex, therefore, an inclusion criterion for study participation if the aim of randomization of the groups were to make sure the sex ratio was equivalent? If no participants were dropped, then was those recruiting participants aware and/or considering the number of males and females that were approached and volunteered for the study?

Answer) Thank you for your comments. Although we did not limit the participants based on age, sex, and race, we used a stratified allocation method based on sex with the purpose of controlling for participant’s sex between the groups. We controlled for sex based on the following reasons: the chest compression depth (CCD) for women rescuers was inferior to those of men rescuers (Hansen et al. PMID: 21629121) because factors such as lower body weight and low level of muscular fitness could decrease CCD (Krikscionaitiene et al. PMID: 22345324, Oh et al. PMID: 27624370, Lopez-Gonzalez et al. PMID: 27344099). The present study was a randomized controlled trial that included a minimal number of participants (n = 40). Therefore, a difference in the sex ratio between the two groups might act as a potential confounder. We have added the reason for using stratified allocation in the methods section.

Although the researcher in charge of collecting data knew the allocated sequence, we did not determine the number of men or women participants prior to starting data collection. We assigned the participants to the groups according to their sex using two randomization lists. The randomization lists were included in the supplementary file (S1 Data set.xlsx). Therefore, the participants also did not know the number of men or women participants and were blinded to the allocation of the group.

5. “Quality,” again, has not been defined for this study. Therefore, when discussing quality, as is in Lines 191-192, in comparison to the quality of chest compression in other studies, it becomes difficult for the reader to understand the comparison/contrast being made. Please define what the study indicates as “quality CPR” and “quality chest compressions”

Answer) Thank you for your comments. The quality of cardiopulmonary resuscitation (CPR) is defined by five key factors required for high quality CPR such as chest compressions of adequate rate, chest compressions of adequate depth, full chest recoil between compressions, minimizing interruptions in chest compressions, and avoiding excessive ventilation (Kleinman et al. PMID: 26472993). However, we agree with the reviewer’s concern that there would be confusions in the interpretation of “quality of CPR”. Therefore, as recommended, we have replaced the term “quality of CPR” with “chest compression depth”.

6. Caution when using the term “significantly.” In lines 198-200, when using the phrase “significantly lower” it eludes to the reader that there was a statistically significant (P < 0.05) outcome where in fact the authors are stating the opposite. For example one can say based on the findings of this study that there was no statistical difference between groups A and B for complete recoil and the proportion of trials where complete recoil was noted was fewer than that required for overall high quality CPR, which is consistent to a previous study. Not that, line 198, “was significantly lower than that required for high quality CPR...”

Answer) Thank you for your comments. We used the term “significantly” to denote extremely low values (the ratio of complete recoil was only 11-12%). However, we agree with the reviewer’s concern. Therefore, we have revised the sentence to state “the proportion of trials with complete recoil were low in both groups”.

7. Lines 201-202, “…the risk of injury might be increased when using the term “at least” contradicts the study’s conclusions and hypothesis. If stating/concluding Group B, with CCD target of “at least”, was better than Group A, CCD target of “approximately”, then by having Lines 201-202 causes doubt and conflicting view point. Be careful on wording in addressing this point.

Answer) Thank you for your comments. We have revised the sentence as “Previous studies have demonstrated that higher survival is associated with deeper CCD in the paediatric population. Considering the negative effects (e.g. decreasing CCD) of defining CCD using the term “approximately”, changes in the terminology used for the guidelines should be considered.”.

8. Based on the third limitation outlined in Lines 208-214, the issue in the discussion and conclusion is that one cannot state that the use of “approximate” vs “at least” being better than the other, as the author has done, without accounting for confounding variables which is the entire phrase of the CCD target. One cannot conclude, based on the study design and data presented, that the differences between the groups were to “approximate” vs “at least” or “one third the anterior-posterior diameter of the chest” vs “4 cm.” It can only be discussed/concluded on the entire phrase itself and not to the segments within the phrase. Would therefore remove the context around Lines 190-191 and Lines 195-197.

Answer) Thank you for your comments. We agree with the reviewer’s concern. Instead of removing the sentence, we have revised the sentences as follows: “The results of the present study showed that the term ‘approximately 4 cm’ could significantly limit the CCD when compared to ‘at least one-third the anterior-posterior diameter of the chest.’ This result suggests that compared to the term ‘at least one-third the anterior-posterior diameter of the chest’, the term ‘approximately 4 cm’ could limit the proportion of trials with complete recoil in addition to CCD. 

Our response to the comments of the Reviewer #2:

1. Lines 110-115. The description of the secondary outcomes is a bit cryptic. For example, how is "correct hand position" defined? What does the number of total compressions measure? When is depth "adequate"? When is recoil "complete"? A detailed description of the outcomes under study is not only a courtesy to the readers (such as me) who are not necessarily experts of resuscitation methods, but it also allows results reproducibility. In addition, in which sense do these variables measure resuscitation performance?

Answer) Thank you for your comments. We agree with the reviewer’s concern. We have revised the sentence for describing secondary outcome variables as “Secondary outcome variables were the ratio of correct hand position around lower half of sternum (correct hand position, %), total compressions (n), the ratio of chest compressions achieving chest compression depth over 38 mm (adequate depth, %), the ratio of complete chest wall recoil within 5 mm from baseline (complete recoil, %), average compression rate (n/min), the ratio of adequate chest compression rate between 100/min to 120/min (adequate rate, %), hands-off time (sec), total ventilations (n), and average ventilation volume (ml).” All parameters were measured by the embedded sensors within the infant manikin (Resusci Baby QCPR, Laerdal Medical, Stavanger, Norway) and transferred to the mobile laptop computer (Simpad SkillReporter, Laerdal Medical). Adequacy of parameters such as depth limit of adequate depth and complete recoil was determined by the manufacturer (Laerdal Medical). However, the range of adequate depth, rate, and location of chest compression were recommended by the cardiopulmonary resuscitation (CPR) guidelines. We evaluated these parameters because these parameters (e.g. adequate depth, adequate rate, complete recoil, minimizing hand-off time, adequate ventilation volume) were recommended by the guideline when defining high quality CPR (Kleinman et al. PMID: 26472993).

2. In the statistical analysis section, the authors correctly say that they use the Chi-square tests or Fisher exact tests to compare proportions. However, in Table 2 differences between proportions are incorrectly tested using the Mann-Whitney U test (at least this is what the note of the table says). Please clarify.

Answer) Thank you for your comments. All variables included in table 2 are continuous variables. Therefore, we compared the variables using two-sided independent sample t-test or Mann-Whitney U test. We used the Chi-square tests or Fisher exact tests for analysing categorical variables in Table 1. Please, check the data in the supplementary file (S1 Data set.xlsx).

3. Lines 89-90: the same manikin was used in all the experiments. Isn't this a limitation of the study? Shouldn't one exploit manikins of different sizes to robustify the results? Please clarify this point.

Answer) Thank you for your comments. Theoretically, the reviewer’s comment is accurate. However, the limitation of using the same manikin model is included in the limitations of the simulation trial. In addition, using various kinds of manikin models or different manikin models can be a confounder. For example, if we conducted a laboratory experiment with different animal models or different experimental tools, the validity of the study results would be doubtful. Although the present study was a human trial, the nature of the methodology was experimental.

Our response to the comments of the Reviewer #3:

1. Their statistical analysis plan appears to be appropriate as well. Though I wish they would have described or mentioned what data was normally distributed or not.

Answer) Thank you for your comments. Instead of describing the normal distribution of data, we showed the method of statistical methods at the bottom of the table. For example, the data analysed by using two-sided independent sample t-test were normally distributed and the data analysed by using Mann-Whitney U test were not normally distributed.

2. It appears that it was only interns or residents that participated in the study. I would have mentioned this in the study, instead of saying medical doctors.

Answer) Thank you for your comments. We have revised the term “medical doctors” to “intern or resident physicians” as recommended. Thank you.

Sincerely yours,

Je Hyeok Oh, MD, PhD

---

## [Editor Report · Decision Letter 1]

3 Mar 2020

PONE-D-19-34142R1

Differences in the performance of resuscitation according to the resuscitation guideline terminology during infant cardiopulmonary resuscitation: “approximately 4 cm” versus “at least one-third the anterior-posterior diameter of the chest”

PLOS ONE

Dear Prof. Oh,

Thank you for submitting your manuscript to PLOS ONE. After careful consideration, we feel that it has merit but does not fully meet PLOS ONE’s publication criteria as it currently stands. Therefore, we invite you to submit a revised version of the manuscript that addresses the points raised during the review process.

Thank you for your edits. The revised manuscript has addressed our concerns. One minor additional question: What specialty of interns and residents were enrolled? We only know that they were certified in BLS. It is possible that the specialty of residents might impact on their CCD. For instance, a pediatric resident (who has more confidence with children) might be more aggressive than an internal medicine resident, leading to deeper compressions by the pediatric residents. If the distribution of different specialities between Group A and B are unequal, this could bias the results.==============================

We would appreciate receiving your revised manuscript by Apr 17 2020 11:59PM. To enhance the reproducibility of your results, we recommend that if applicable you deposit your laboratory protocols in protocols.io, where a protocol can be assigned its own identifier (DOI) such that it can be cited independently in the future. For instructions see: http://journals.plos.org/plosone/s/submission-guidelines#loc-laboratory-protocols

We look forward to receiving your revised manuscript.

Kind regards,

Kevin Ching, M.D.

Academic Editor

PLOS ONE

---

## [Author Response · Author response to Decision Letter 1]

4 Mar 2020

4 March 2020

Kevin Ching, MD

Academic Editor

PLOS ONE

Dear Editor, 

I wish to re-submit the manuscript titled “Differences in the performance of resuscitation according to the resuscitation guideline terminology during infant cardiopulmonary resuscitation: “approximately 4 cm” versus “at least one-third the anterior-posterior diameter of the chest”.” The manuscript ID is PONE-D-19-34142_R1.

We thank you and the reviewers for your thoughtful suggestions and insights. The manuscript has benefited from these insightful suggestions. I look forward to working with you and the reviewers to move this manuscript closer to publication in PLOS ONE. The manuscript has been rechecked and the necessary changes have been made in accordance with the reviewers’ suggestions. Thank you for your consideration. I look forward to hearing from you.

Sincerely,

Je Hyeok Oh, MD, PhD

Department of Emergency Medicine,

Chung-Ang University College of Medicine,

84 Heukseok-ro, Dongjak-gu, Seoul 06974, Republic of Korea

Tel. no.: +82 2 6299 1820

Fax no.: +82 2 6264 1119

Email address: jehyeokoh@cau.ac.kr 

Our response to the comments of the Reviewer #1:

1. Thank you for your edits. The revised manuscript has addressed our concerns. One minor additional question: What specialty of interns and residents were enrolled? We only know that they were certified in BLS. It is possible that the specialty of residents might impact on their CCD. For instance, a pediatric resident (who has more confidence with children) might be more aggressive than an internal medicine resident, leading to deeper compressions by the pediatric residents. If the distribution of different specialities between Group A and B are unequal, this could bias the results.

Answer) Thank you for your valuable comments. We have revised the manuscript according to the reviewer’s recommendation. In detail, there were no specialities in the intern physician. In South Korea, intern physicians rotate all clinical fields (specialities) for one year before starting the course of resident physician. Therefore, we have compared whether there was a difference in the specialities of the resident physician according to the groups. As a result, there were no differences in the distribution of specialities between Group A and B. We have added the results in the table 1.

---

## [Editor Report · Decision Letter 2]

6 Mar 2020

Differences in the performance of resuscitation according to the resuscitation guideline terminology during infant cardiopulmonary resuscitation: “approximately 4 cm” versus “at least one-third the anterior-posterior diameter of the chest”

PONE-D-19-34142R2

Dear Dr. Oh,

We are pleased to inform you that your manuscript has been judged scientifically suitable for publication and will be formally accepted for publication once it complies with all outstanding technical requirements.

With kind regards,

Kevin Ching, M.D.

Academic Editor

PLOS ONE
---

## [Editor Report · Acceptance letter]

11 Mar 2020

PONE-D-19-34142R2 

Differences in the performance of resuscitation according to the resuscitation guideline terminology during infant cardiopulmonary resuscitation: “approximately 4 cm” versus “at least one-third the anterior-posterior diameter of the chest” 

Dear Dr. Oh:

I am pleased to inform you that your manuscript has been deemed suitable for publication in PLOS ONE. Congratulations! Your manuscript is now with our production department. 

With kind regards,

on behalf of

Dr. Kevin Ching 

Academic Editor

PLOS ONE